# Immunocheckpoint Inhibitors in Microsatellite-Stable or Proficient Mismatch Repair Metastatic Colorectal Cancer: Are We Entering a New Era?

**DOI:** 10.3390/cancers15215189

**Published:** 2023-10-28

**Authors:** Laura Matteucci, Alessandro Bittoni, Graziana Gallo, Laura Ridolfi, Alessandro Passardi

**Affiliations:** 1Department of Medical Oncology, IRCCS Istituto Romagnolo per lo Studio dei Tumori (IRST) “Dino Amadori”, 47014 Meldola, Italy; 2Operative Unit of Pathologic Anatomy, Azienda USL della Romagna, “Maurizio Bufalini” Hospital, 47521 Cesena, Italy

**Keywords:** metastatic colorectal cancer, microsatellite stability, proficient mismatch repair, immunotherapy, immunocheckpoint inhibitors

## Abstract

**Simple Summary:**

The majority of metastatic colorectal cancer cases are mismatch-repair-proficient and microsatellite-stable, and unfortunately this condition is associated with an inherent resistance to Immune Checkpoint Inhibitors. However, several trials are investigating the right way to overcome resistance in these tumors, so as to expand the application scope of immunotherapy. Future perspectives include mainly a combination of immune checkpoint inhibitors with chemotherapy and bevacizumab or cetuximab; sequential treatment with Temozolomide in O6-methylguanine-DNA methyltransferase (MGMT)-methylated tumors; validation of Immunohistochemical biomarkers of response, such as tumor mutational burden or Immunoscore.

**Abstract:**

Colorectal cancer (CRC) is the third most frequent cancer and the second leading cause of cancer-related deaths in Europe. About 5% of metastatic CRC (mCRC) are characterized by high microsatellite instability (MSI) due to a deficient DNA mismatch repair (dMMR), and this condition has been related to a high sensitivity to immunotherapy, in particular to the Immune Checkpoint Inhibitors (ICIs). In fact, in MSI-H or dMMR mCRC, treatment with ICIs induced remarkable response rates and prolonged survival. However, the majority of mCRC cases are mismatch-repair-proficient (pMMR) and microsatellite-stable (MSS), and unfortunately these conditions involve resistance to ICIs. This review aims to provide an overview of the strategies implemented to overcome ICI resistance and/or define subgroups of patients with MSS or dMMR mCRC who may benefit from immunotherapy.

## 1. Introduction

Colorectal cancer (CRC) is the third most frequent cancer in adults and the second leading cause of cancer-related deaths in Europe [1,2,3,4]. Approximately 25% of CRC patients have metastatic disease at diagnosis, and among those presenting with early-stage disease, about 50% will develop metastases. Metastatic colorectal cancer (mCRC) remains an incurable disease characterized by a poor prognosis, although improvements in knowledge and treatments have made it possible to increase the average survival from a few months to almost 3 years [5,6]. Immune checkpoint inhibitors (ICIs) are monoclonal antibodies that act by blocking checkpoint proteins on the surface of immune cells (such as cytotoxic T-lymphocyte antigen-4, CTLA-4, or programmed death-1, PD-1) from binding with their partner proteins on the tumor cells. This prevents the “off” signal from being sent, allowing the T cells to kill cancer cells. These drugs have undergone rapid development in recent times, guaranteeing excellent results in numerous solid neoplasms, including mCRC [7,8]. In particular, the efficacy of ICIs has been demonstrated in mCRC that is mismatch-repair-deficient (dMMR) or shows high levels of microsatellite instability (MSI-H) [9], whereas it hasn’t been found in tumors that are mismatch-repair-proficient (pMMR) and microsatellite-stable (MSS) [8]. Unfortunately, MSI/dMMR tumors are only about 5% of all mCRCs [10]. For this reason, in recent years, numerous attempts have been made to better understand the mechanisms of resistance to ICIs by MSS tumors. Moreover, several trials are currently exploring the possibility of overcoming ICI resistance and/or defining subgroups of patients with MSS or dMMR mCRC who might benefit from immunotherapy. At present, the main strategy is to combine immunotherapy with other treatments, such as chemotherapy, target therapy, and radiotherapy, and preliminary data are encouraging (Figure 1). In this article, we aim to review these studies and the main evidence in this field.

## 2. Clinical Trials of ICI in MSI/dMMR mCRC

Several clinical trials have analyzed the efficacy of ICIs in MS-HI/dMMR mCRC (Table 1). Pembrolizumab, a humanized monoclonal immunoglobulin (Ig) G4 antibody directed against the human cell surface receptor PD-1, was investigated in the phase II trial KEYNOTE-016 in patients with both MSI-H and MSS mCRC, showing an objective response rate (ORR) of 50% and 0%, respectively. Notably, in the MSI mCRC cohort, the 2-year progression-free survival (PFS) and overall survival (OS) rates were 61% and 66%, respectively [11]. These results lead in 2017 to the approval from the Food and Drug Administration (FDA) and European Medicines Agency (EMA) of Pembrolizumab for pretreated patients with MSI-H mCRC. Subsequently, the phase III KEYNOTE-177 trial was designed to evaluate first-line pembrolizumab compared to a standard 5-fluorouracil-based chemotherapy. Trial results were indicative of the superiority of immunotherapy over chemotherapy in terms of PFS (16.5 vs. 8.2 months in pembrolizumab and standard treatment arms, respectively; hazard ratio (HR): 0.6; *p* = 0.0002) [12]. A statistically significant advantage in terms of OS was not observed in the experimental arm—probably due to the masking effect of the crossover—in 60% of the intention-to-treat (ITT) population [13]. These results lead in 2020 to FDA and EMA approval of pembrolizumab as standard first-line treatment for patients with MSI-H mCRC. 

The efficacy of Nivolumab, a fully human IgG4 anti-PD-1 monoclonal antibody, alone or in combination with Ipilimumab, a fully human monoclonal IgG1κ antibody against the CTLA-4, was investigated in the CheckMate 142 phase II trial in both pretreated and treatment-naïve MSI-H mCRC patients. In pretreated patients, nivolumab as a single agent induced an ORR of 31.1% and a disease control rate (DCR) of 69%. Median PFS was 14.3 months, and median OS was not reached. In pretreated patients assigned to the Ipilimumab–Nivolumab combination, an ORR of 65%, including 13% of complete responses (CR), and a DCR of 85% were shown. The median PFS and OS were not reached, while 48-month PFS and OS rates were 53% and 71%, respectively [14]. In naive patients, the combination of nivolumab plus ipilimumab was associated with a similar ORR (69%, with CR in 13% of patients), and DCR (84%). Notably, median PFS and median OS were not reached, 24-month OS was 74% and 24-month PFS was 79% [15]. In July 2017, nivolumab alone or in combination with ipilimumab was approved by the FDA as standard treatment for patients with MSI-H mCRC after failure of at least one line of chemotherapy. The CheckMate 8HW (NCT04008030) is an ongoing international, multicenter, open-label, randomized, phase 3 study designed to compare the efficacy and safety of nivolumab plus ipilimumab to chemotherapy (investigator’s choice) or single-agent nivolumab in patients with MSI-H/dMMR mCRC [16]. The dual primary endpoints are PFS for nivolumab plus ipilimumab versus nivolumab across all lines, and nivolumab plus ipilimumab versus chemotherapy in the first line of treatment. The results of the CheckMate 8HW trial are eagerly awaited, with a high probability of extending the indication of the combination of Nivolumab plus Ipilimumab to the first-line setting as well.

## 3. Combination of ICIs and Chemotherapy

Preclinical data suggest that the combination of chemotherapy and immunotherapy may overcome the mechanisms of primary resistance to ICI monotherapy in pMMR/MSS mCRC [17,18,19,20,21,22,23,24]. For this reason, several trials are investigating ICIs combined with chemotherapy alone or regimens containing Bevacizumab or anti-epidermal growth factor receptor (EGFR) monoclonal antibodies (Table 2).

Phase I trial: a trial aimed to characterize the safety, tolerability, and pharmacokinetics of an investigational drug and identify a recommended dose and regimen for future studies; Phase II trial: a trial aimed to assess the activity and safety of an investigational drug or combination in a particular indication; Phase III trial: a randomized trial aimed to assess efficacy of an investigational drug compared to standard treatment on large patient groups.

### 3.1. ICIs in Combination with Chemotherapy Alone

Among the most frequently used chemotherapeutic agents in the treatment of mCRC, 5-fluorouracil (5-FU) and oxaliplatin may have an immunostimulatory effect, whereas irinotecan seems to enhance immunosuppression. In particular, 5-FU causes apoptosis of myeloid-derived suppressor cells (MDSCs) favoring tumor infiltration by cytotoxic T lymphocytes, whereas Irinotecan blocks MDSC apoptosis and myeloid cell differentiation, increasing MDSC immunosuppressive features [25,26]. While Oxaliplatin kills colon cancer cells through the induction of DNA damage, it seems to induce the overexpression of tumor-associated antigens and the downregulation of programmed death-ligand 1 (PD-L1) expression [27,28,29,30]. This results in the activation of CD4 T helper cells in antitumor immunity. Interestingly, in an immune-resistant mouse model of colon cancer, Oxaliplatin in combination with ICIs showed an antitumor response [31]. 

All these findings provide the rationale for identifying the FOLFOX (Folinic acid, 5FU plus Oxaliplatin) regimen, as the basis of new combination treatment strategies to enhance the efficacy of ICIs in mCRC (Table 2). A phase II single-arm study (NCT02375672) evaluated the combination of FOLFOX regimen and Pembrolizumab in patients with untreated mCRC. Of 30 patients included, three presented dMMR, 22 pMMR, and five unknown MMR status. The combination FOLFOX/Pembrolizumab achieved a noteworthy ORR of 53% with a DCR of 100% at 8 weeks, suggesting a clinical activity in patients with pMMR untreated mCRC [32]. Moreover, FOLFOX in addition to durvalumab and tremelimumab as induction therapy (six cycles), followed by maintenance therapy with durvalumab, was assessed in the phase Ib/II MEDETREME trial (NCT03202758). The trial showed a promising 6-month PFS rate (primary endpoint of the study) of 63.2% and an ORR of 61%, with a DCR of 89%, in patients with previously untreated RAS-mutated mCRC. Interestingly, of 57 patients enrolled only three (5%) were MSI-H. Results of the one-year follow-up showed PFS rates similar to those achieved with a standard first-line regimen, with the advantage of administering only 3 months of induction chemotherapy [33,34]. 

In preclinical studies, trifluridine/tipiracil (FTD/TPI) showed immunomodulatory properties; however, the single-arm and safety lead-in phase 2 trial NCT02860546 failed to show an antitumor activity of FTD/TPI in combination with nivolumab in patients with refractory MSS mCRC. In fact, although safety data in this population indicated tolerability and feasibility of this combination, no patient achieved a tumor response [35].

### 3.2. ICIs in Combination with Chemotherapy and Bevacizumab

Bevacizumab is an antiangiogenic monoclonal antibody used in combination with chemotherapy in the first- and second-line treatments of mCRC. The inhibition of the Vascular Endothelial Growth Factor (VEGF) pathway, through the binding of Bevacizumab to VEGF-A, has both an antitumor effect, by reducing neoangiogenesis, and an immunostimulatory effect, by promoting dendritic cell (DC) maturation, proliferation and cytotoxic activity of T cells, and reducing the expansion of Tregs and MDSCs [36,37,38,39,40,41]. On the basis of this evidence, the combination of chemotherapy with Bevacizumab and ICIs has been evaluated in several clinical trials with conflicting results (Table 2). 

In the MODUL study (NCT02291289), the addition of the anti PD-L1 monoclonal antibody Atezolizumab to maintenance therapy with fluoropyrimidine and bevacizumab, after induction therapy with FOLFOX and Bevacizumab for six or eight cycles, did not improve PFS in patients with pMMR and BRAF wild-type mCRC (cohort 2). Indeed, the PFS benefit, the primary efficacy endpoint, was not achieved either at a median follow-up of 10.5 months or at a median follow-up of 20.3 months. There were also no differences in terms of OS, ORR, and DCR [42,43,44]. 

In the first-line setting, the combination of a triplet chemotherapy regimen and immunotherapy was recently evaluated in the Phase II/III ATEZOTRIBE study (NCT03721653) in which patients were randomized to receive an induction treatment of eight cycles of FOLFOXIRI plus Bevacizumab with Atezolizumab followed by maintenance with 5FU plus bevacizumab and atezolizumab (experimental arm), or induction of eight cycles of FOLFOXIRI plus Bevacizumab followed by maintenance with 5FU plus bevacizumab (standard arm). The experimental arm resulted in a significant increase in PFS, primary end point, of 13.1 months, compared with 11.5 months for the control group (*p* = 0.012). Approximately 91.3% of the ITT population had MSS status; in this subgroup the PFS with the addition of atezolizumab was 12.9 months versus 11.4 months in the control group (*p* = 0.071) [45,46]. Similarly, the Phase II NIVACOR study (NCT04072198) evaluated the efficacy of FOLFOXIRI and Bevacizumab with Nivolumab versus FOLFOXIRI and Bevacizumab as first line followed by maintenance with Bevacizumab, associated with Nivolumab in the experimental arm, in untreated patients with RAS/BRAF-mutated mCRC, regardless of microsatellite status [47]. From October 2019 to March 2021, 73 patients were enrolled in nine Italian centers, of whom 10 were MSI-H (16.1%), 52 (83.9%) MSS, and 11 not assessed. The ORR was 76.7%, with 7 (9.6%) CR, and DCR was 97.3%. The median and 12-month PFS were 10.1 months (95% CI: 9.4-NE) and 53.4%, respectively. Notably, in the subgroup of MSS patients, the ORR was 78.9% with a DCR of 96.2%, and the median PFS was 9.8 (95% CI: 8.18–15.24) months. In the Phase II/III CheckMate9X8 study (NCT03414983) FOLFOX combined with bevacizumab and nivolumab versus FOLFOX and bevacizumab was evaluated as first-line treatment in patients with mCRC. Although the primary endpoint of PFS was not met, nivolumab in combination with FOLFOX and bevacizumab showed higher 15-month and 18-month PFS rates, a higher response rate and longer-lasting responses compared to the standard of care, with an acceptable safety profile [48]. 

3B-FOLFOX is a Phase I/II study (NCT05627635) designed to explore the safety and clinical efficacy of the combination FOLFOX plus bevacizumab with Botensilimab (anti PD-1) and Balstilimab (anti CTLA-4) in patients with MSS mCRC, regardless of RAS/BRAF mutational status from first through third line of treatment [49]. Similarly, the ongoing phase Ib/II COLUMBIA-1 trial (NCT04068610) aims to compare FOLFOX and bevacizumab in combination with durvalumab and the anti-CD73 monoclonal antibody Oleclumab vs. standard of care FOLFOX plus Bevacizumab as first-line treatment [50]. Furthermore, POCHI is an ongoing multicenter and single-arm phase II trial (NCT04262687) to assess the efficacy of Pembrolizumab with CAPOX and Bevacizumab as first-line treatment of MSS/pMMR mCRC with a high immune infiltrate, evaluated on resected primary colorectal cancer [51]. In the BACCI trial (NCT02873195), the addition of atezolizumab to capecitabine and bevacizumab did not provide a clinically meaningful increase in progression-free survival (PFS) compared with placebo (median PFS 4.4 months vs. 3.6 months) among patients with refractory mCRC. Also, median OS was similar in the investigational and placebo groups (10.3 and 10.2 months, respectively). Of 128 patients included, 110 (89.4%; 69 in the investigational group and 41 in the placebo group) had MSS/pMMR disease [52,53]. Lastly, the aim of the ongoing phase 2 trial NCT05314101 is to evaluate the clinical efficacy and safety of TAS-102 combined with bevacizumab and tislelizumab in third-line or above treatment in patients with mCRC and liver metastasis [54].

Results from ongoing trials are eagerly awaited to further strengthen the rationale for the combination of bevacizumab and/or oxaliplatin-based chemotherapy and immunotherapy.

### 3.3. ICIs in Combination with Chemotherapy and Anti-EGFR Agents

Anti-EGFR monoclonal antibodies in combination with chemotherapy or alone represent a standard treatment for RAS and BRAF wild-type mCRC [55,56,57,58,59]. Cetuximab is a chimeric IgG1 monoclonal antibody that acts through antibody-dependent cellular toxicity (ADCC) promoting the expression of major histocompatibility complex (MHC) class II molecules on DCs [60,61,62]. Panitumumab is a fully human IgG2 monoclonal anti-EGFR antibody that did not show ability in activating innate and adaptive immune cells against tumor cells. Also, in the combination chemotherapy setting, the immunogenity of Panitumumab was infrequent and similar to that observed in the monotherapy setting [63,64,65]. 

In the first-line setting, the role of avelumab (a PD-L1-blocking human IgG1 lambda monoclonal antibody) in addition to FOLFOX and cetuximab in patients with RAS and BRAF wild-type mCRC, regardless of the microsatellite status, has been investigated in the phase II AVETUX clinical trial (NCT03174405). Among 43 enrolled patients, 40 (93.0%) were MSS and three (7%) were MSI. Four patients (9.3%) were excluded from the analysis after enrollment due to the identification of low-frequency KRAS or BRAF mutations at the central tissue review. Although the primary endpoint of increasing the 12-month PFS rate from 40% to 57% was not met, as 12-month PFS was 40%, avelumab in combination with FOLFOX showed a promising ORR of 79.5% and a DCR of 92.3%. The combination was well-tolerated without unexpected adverse events over standard FOLFOX and cetuximab [66]. The FOLFIRI regimen plus cetuximab in combination with Avelumab, followed by maintenance with Avelumab, is under investigation in the FIRE-6 phase II trial (NCT05217069) [67].

The Phase II VOLFI and MACBETH studies demonstrated that the chemotherapy triplet regimen FOLFOXIRI in combination with an antiEGFR agent (Panitumumab and Cetuximab, respectively) as first-line treatment improves ORR and the rate of secondary resection of metastases, albeit without a significant increase of the PFS and OS, in RAS and BRAF wild-type mCRC [68,69]. On this basis the Phase II AVETRIC study (NCT04513951) was designed to evaluate efficacy and safety of first-line FOLFOXIRI and cetuximab with Avelumab for up to 12 cycles, followed by maintenance with 5-FU and cetuximab associated to Avelumab, in treatment-naïve patients with wild-type RAS and BRAF, MSI, or MSS mCRC [70]. 62 patients were enrolled in 16 Italian centers. The primary endpoint was met, as mPFS was 14.1 months (90% CI: 12.0–16.7, Brookmeyer-Crowley test *p* < 0.001). ORR and DCR were 82% and 98%, respectively, and R0 resection rate was 21% (27% in the liver-only subgroup of patients). 

In the chemorefractory setting, the rechallenge strategy with anti-EGFR agents was demonstrated to be active in patients with RAS and BRAF wild-type mCRC after initial response to anti-EGFR-based first-line therapy [71]. Based on this evidence, the phase II single-arm CAVE trial (NCT04561336) was conducted to assess the clinical efficacy of cetuximab in combination with avelumab as third-line rechallenge therapy in patients with RAS wt mCRC who had a CR or PR to first-line chemotherapy plus anti-EGFR drugs and failed second-line chemotherapy. No selection was made on the basis of microsatellite status and approximately 92% of the 77 enrolled patients had MSS tumors. The trial achieved its primary endpoint, with a median OS of 11.6 months (95% CI: 8.4–14.8 months), and a median PFS of 3.6 months (95% CI: 3.2–4.1 months). Moreover, median OS and PFS were longer in patients with RAS and BRAF wild-type circulating tumor DNA (ctDNA) (17.3 vs. 10.4 months, *p* = 0.02, and 4.1 vs. 3.0 months, *p* = 0.004, respectively) detected at baseline [72,73,74,75,76]. These preliminary results are further investigated in the ongoing randomized phase II CAVE2 study, whose purpose is to compare cetuximab in association with avelumab versus cetuximab alone as rechallenge strategy [77,78].

Irinotecan and Cetuximab, one of the standard rechallenge regimen, in combination with avelumab was evaluated in the single-arm phase II AVETUXIRI (NCT03608046) study among patients with BRAF wt MSS refractory mCRC, regardless of RAS mutational status. Patients with RAS wt mCRC were enrolled into cohort A, and those with RAS mutations were enrolled into cohort B. At the interim analysis, 23 patients had been included, 10 in the cohort A and 13 in the cohort B. The trial met its primary efficacy endpoint for RAS-wt patients as the immune-related ORR, according to immune RECIST (irRECIST1.1), was 30% in cohort A, whereas no CR or PR were shown in cohort B. The safety profile was favourable, without unexpected adverse events. Disease control rate was 60.0% in cohort A and 61.5% in cohort B. Six-month PFS and 12-month OS rates were higher in cohort A; 40.0% and 50.0% (cohort A) and 38.5% and 46.2% (cohort B), respectively. Also, median PFS was longer among RAS wild-type patients (4.2 months versus 3.8 months in RAS-mutated patients), whereas median OS was 12.7 months in cohort A and 14.0 months in cohort B. Furthermore, an immunoscore was generated based on the density of CD3+ (T cells) and CD8+ (cytotoxic) detected in the metastases biopsies. Patients with a high immunoscore had a significantly higher tumor shrinkage (OR = 18.67 *p* = 0.019) and longer median PFS (6.9 vs. 3.4 months; HR = 0.16, *p* = 0.002) and median OS (13.7 vs. 7.9 months, HR = 0.26, *p* = 0.009), regardless of RAS mutations [79,80,81].

NCT03442569 was a multicenter, single-arm, phase II clinical trial of panitumumab, ipilimumab, and nivolumab in 49 pretreated RAS/BRAF WT, MSS mCRCs. The trial met its primary endpoint, since the 12-week ORR was 35% (95% CI: 21–48; *n* = 17 responses). Median PFS was 5.7 months (95% CI: 5.5–7.9), comparing favorably to expected PFS for anti-EGFR monotherapy in RAS wild-type patients. Overall, trial results suggested an activity of the ICI/anti-EGFR combination in refractory RAS/BRAF WT and MSS mCRC [82].

### 3.4. ICIs in Combination with Temozolomide

Temozolomide is an oral alkylating agent approved for patients with glioblastoma, and the efficacy of the drug in this disease is related to a validated predictive biomarker; the O6-methylguanine-DNA methyltransferase (MGMT) promoter methylation. MGMT methylation is found in about 40% of CRC and a few studies have shown a modest activity of temozolomide in MGMT-methylated mCRCs [83]. Interestingly, acquired resistance to temozolomide may be associated with the onset of inactivating mutations in MMR genes, such as MSH6, and increased tumor mutational burden (TMB). These observations lead to the design of studies evaluating a sequence of temozolomide priming followed by immunotherapy with anti-PD-1 agents in MGMT-methylated mCRCs. 

In the MAYA trial [84], 135 pretreated MGMT-methylated pMMRmCRC patients were enrolled and treated with two priming cycles of oral temozolomide, followed, in the absence of progression, by its combination with ipilimumab 1 mg/kg once every 8 weeks and nivolumab 480 mg. After the first treatment part, 33 patients (24%) achieved disease control and started immunotherapy with ICIs. The study met its primary endpoint with an 8-month PFS rate of 36% in patients who started the second treatment part. Median PFS and OS were 7.0 and 18.4 months, respectively, confirming how this strategy may achieve durable disease control in treatment-refractory mCRC patients. 

Another study, the ARETHUSA trial [85], is currently evaluating a similar treatment strategy in MGMT-methylated, RAS mutant, pMMR mCRC. In the priming phase of the trial, patients are treated with temozolomide until disease progression or unacceptable toxicity. At the time of disease progression, a biopsy is performed to assess TMB, and patients with tumor mutational load ≥20 mutations/MB proceed to the immunotherapy phase and receive pembrolizumab. Preliminary results of the trial have been recently published and the experimental treatment showed promising activity with a DCR of 67% in the first six patients treated with pembrolizumab. Interestingly, the analysis of tissue biopsies and ctDNA in the first 21 enrolled patients confirmed the emergence of a distinct mutational signature and increased TMB after treatment with temozolomide.

These data provide proof-of-concept for the induction of hypermutability through the use of temozolomide as a potential pathway for enabling a response to immunotherapy in pMMR CRC. Notably, in both trials, only 5% of initially screened patients were eligible for ICIs treatment because of molecular selection and temozolomide-driven selection. 

## 4. ICIs in Combination with Target Therapy

The Mitogen-Activated Protein Kinases (MAPK) pathway is crucial in pathogenesis of CRC and represents the downstream cascade of multiple growth factor receptors including EGFR. Interestingly, recent data have shown that activating mutations in the RAS/BRAF/MEK/ERK pathway are associated with an immunosuppressive phenotype. In particular, it has been demonstrated that the MPAK pathway may control PD-L1 and CTLA-4 expression [86] but may also impact on the tumor microenvironment (TME), promoting an immunosuppressive stroma through the secretion of cytokines and growth factors [87]. These observations provide the biological rationale for strategies of combination treatment including ICIs and inhibitors of the MAPK pathway in pMMR CRCs. The principal trials investigating ICIs in combination with targeted agents are summarized in Table 3.

### 4.1. ICIs in Combination with KRAS Inhibitors 

KRAS mutations, detected in about 30–50% of CRCs, are among the signature mutations in CRC. Besides their role as predictive factors for anti-EGFR treatment, KRAS mutation has recently become a target in CRC treatment. In fact, the introduction of KRAS inhibitors, specifically for KRAS G12C mutations such as Sotorasib (AMG510) and Adagrasib (MRTX849), provided a novel treatment option for this subgroup of patients. Data from phase II trials showed promising results with KRAS G12C inhibitors, particularly in terms of disease DCR (about 80%) [88], in pretreated patients. Furthermore, data on the combination treatment with anti-EGFR antibodies are even more encouraging, with ORR reaching 46% for the combination of adagrasib and cetuxumab [89]. 

Interestingly, KRAS mutations have been shown to induce an immunosuppressive phenotype in CRC through different means, including downregulation of MHC class I molecules and conversion of CD4+ cells to Tregs [90], and recent preclinical studies demonstrated an immunomodulatory role for KRAS inhibitors. In particular, Sotorasib was able to increase infiltration of T cells, primarily CD8+, DCs, and macrophages in KRAS G12C-mutated murine models, promoting a pro-inflammatory TME [91]. On the basis of this evidence, Sotorasib in combination with anti-PD-1 therapy is currently being evaluated in patients with KRAS G12C-mutated mCRC in the phase Ib Codebreak101 trial [92]. Results of this trial, and other similar studies, will clarify the safety and feasibility of this combination.

### 4.2. ICIs in Combination with BRAF and MEK Inhibitors

BRAF V600E mutation occurs in about 8–10% of CRCs and represents a marker of poor prognosis in the metastatic setting. Simultaneous inhibition of BRAF and EGFR is an effective treatment strategy in BRAF mutant mCRC, as shown in the BEACON trial; a phase III trial assessing two different combinations (encorafenib + binimetinib + cetuximab and encorafenib + cetuximab) in pretreated patients. The study established encorafenib plus cetuximab as a standard treatment in BRAF V600E mutant mCRC which had progressed after one or two previous regimens [93]. 

About 15–20% of BRAF mutant CRCs are also dMMR, whereas the majority presents as pMMR. The subgroup analysis of BEACON trial did not show any significant interaction between MSI status and encorafenib + cetuximab +/− binimetinib, but notably the percentage of MSI patients included in the study was low (8%). On the other hand, initial clinical trials of BRAF/MEK/EGFR inhibition showed that durable responses to treatment may be restricted to BRAF mutant and dMMR patients, suggesting the possibility that the BRAF pathway inhibition may enhance the immune response in BRAF V600E-mutated CRCs [94].

On this basis, a recent phase I/II trial assessed the triplet encorafenib, cetuximab and nivolumab in BRAFV600E pretreated pMMR mCRC. Preliminary results, presented at ASCO Gastrointestinal Cancers Symposium 2022, were encouraging with an ORR of 50%, a median PFS of 7.4 months, and a median OS of 15.1 months, comparing well with the results of the BEACON trial [95]. A randomized phase II trial (NCT05308446) is currently comparing this combination with encorafenib plus cetuximab in this setting to confirm the results.

Interestingly, preclinical models suggest that BRAF and MEK inhibition may synergize with ICI activity. In particular, both dabrafenib and trametinib, alone or in combination, increased tumor infiltration of lymphocytes, and enhanced the antitumor effect with adoptive T-cell immunotherapy in murine models [96]. Two different combinations are currently being evaluated in patients affected by BRAF V600E-mutated pMMR CRC. A phase I/II trial (NCT04044430) is evaluating combination of encorafenib + binimetinib + nivolumab, while a phase II trial (NCT03668431) assesses dabrafenib + trametinib + spartalizumab, an anti-PD-1 drug. Preliminary data of the latter study, including 37 patients, demonstrated promising activity with an ORR of 24.3% [97]. In the same study, single-cell RNA sequencing of 23 paired pretreatment and day 15 on-treatment tumor biopsies revealed induction of tumor-cell-intrinsic immune programs (such as types I and II IFN response, antigen-presenting genes, and T-cell-recruiting chemokines) and a more complete MAPK inhibition in patients with better clinical outcome.

Other trials are evaluating the combination of MEK inhibitors and ICIs in mCRC. A phase Ib study investigated the safety and clinical activity of a combination of cobimetinib and atezolizumab in patients with solid tumors, including mCRC, melanoma, and non-small-cell lung cancer. Objective responses were observed in seven of 84 patients (8%) with mCRC, of whom, six patients had MSS disease [98]. Promising results were also observed in a different phase Ib trial which evaluated the same combination in 24 patients with pretreated mCRC. The ORR was 17% and three out of four responders were pMMR [99]. In the phase III trial IMblaze370, 363 patients with pretreated mCRC, mostly with MSS tumors, were randomized to receive atezolizumab alone or in combination with cobimetinib or regorafenib. The primary endpoint of the study was OS. Unfortunately, the study did not confirm the encouraging results shown in the previous trials. In particular, the study demonstrated a median OS of 8.9 months in the combination arm, 7.1 months in the atezolizumab-alone arm, and 8.5 months in the regorafenib arm. No differences in PFS and OS were found in the three arms, even in the clinical and biomarker subgroup analyses, including patients with extended *RAS* mutation or high PD-L1 expression [100]. 

### 4.3. ICIs in Combination with PIK3CA/AKT/mTOR Inhibitors

The PIK3CA/AKT/mTOR signalling pathway plays an important role in cancer cell survival, angiogenesis, and metastasis in CRC [101]. Activating mutations of PI3KCA are quite common in CRCs (7–30%). Nevertheless, the prognostic and predictive role of this mutation has not been fully clarified, in part due to the frequent co-occurrence of KRAS mutations. Recently, the inhibition of the PI3k/Akt/mTOR pathway has become a promising therapeutic strategy in CRC patients with some encouraging preliminary results [102]. Recent studies demonstrated how inhibition of PI3Kα/δ sub-units may enhance the CD8+ T-cell activity and decrease the number of suppressive T-reg, in tumor microenvironment [103]. On this basis, a phase I/II clinical trial is currently investigating the combination of Nivolumab + Copanlisib, a PIK3CA inhibitor, in pMMR mCRC patients [104]. 

### 4.4. ICIs in Combination with Multitarget Tyrosine Kinase Inhibitors (TKIs)

Regorafenib is a multitarget TKI currently approved as salvage third-line treatment in mCRC. Beyond its antiangiogenic activity, regorafenib is able to reduce tumor-associated macrophages (TAMs) through the inhibition of the colony-stimulating factor 1 (CSF1) receptor, as demonstrated in tumor models [105]. Considering the role of TAMs as an immunosuppressive component in TME, the concomitant use of regorafenib might represent a strategy to improve immunotherapy efficacy in pMMR CRC. The REGONIVO trial, a phase Ib Japanese study on metastatic gastric cancer and mCRC patients, evaluated the combination of regorafenib and nivolumab [106]. The study enrolled 25 pretreated mCRC patients—24 with pMMR and 1 with dMMR tumors—and showed encouraging results, with an ORR of 33% and a median PFS of 7.9 months in the pMMR cohort. Interestingly, patients with liver metastasis had lower response rates when compared to patients with lung metastasis (15% vs. 50%, respectively). An exploratory biomarker analysis demonstrated no relationship between PD-L1 expression or TMB and outcomes. A phase II study evaluating the same combination of regorafenib and nivolumab in a cohort of 70 pMMR mCRC patient showed less favourable results with an ORR of 7% and a median PFS of 1.8 months [107]. These differences in outcome may be explained by baseline characteristics of the study populations, including the performance status, the tumor sidedness, and the presence of liver metastases, that were, in general, more favourable in the Japanese study. The study confirmed the different effect of this regimen in patients with and without liver metastases. In particular, all five responders have no liver metastases, and ORR among patients without liver metastasis was 22%. Similar disappointing results were found in a study evaluating regorafenib in association with pembrolizumab that demonstrated a median PFS and OS of 2.0 and 10.9 months, respectively, with no objective responses in pretreated mCRC patients [108]. Also, the combination of regorafenib and avelumab was explored in the REGOMUNE trial, a phase II trial including 48 pretreated pMMR mCRCs. SD was achieved in 23 patients (53.5%), and PD in 17 patients (39.5%). The median PFS and OS were 3.6 months and 10.8 months, respectively [109]. 

More encouraging results were obtained in the phase Ib CAMILLA trial: a study evaluating the combination of cabozantinib and durvalumab in patients with pMMR gastrointestinal malignancies, including mCRC. Overall, the ORR was 30% with a DCR of 83.3%, whereas in the mCRC cohort (*n* = 17) the ORR and DCR were 23.5% and 88.2% and the median PFS and OS were 4.6 and 9.6 months, respectively [110]. The biomarker analysis showed that PD-L1 combined positive score (CPS), tumor CD68 and CD4 protein levels (representing cell surface protein markers for TAMs), and tumor-infiltrating CD4 T cells could be potential predictive markers for cabozantinib plus durvalumab activity in this setting. The combination of a different TKI, lenvatinib, with pembrolizumab was assessed in the LEAP-005 trial in the same patient population, showing an ORR of 22% and an mPFS of 2.3 months, with a manageable safety profile, even though a toxic death for gastrointestinal perforation was reported [111].

### 4.5. ICIs in Combination with PARP Inhibitors

Poly (ADP-ribose) polymerase (PARP) inhibitors (PARPi) are oral drugs which interfere with the DNA repair machinery, particularly in tumors with existing defects in double-strand breaks repair, and induce synthetic lethality [112]. Tumor cells harboring mutations in DNA damage response (DDR) key genes, belonging to non-homologous end-joining (NHEJ) and homologous recombination (HR) pathways, are more sensitive to PARP inhibitors action. PARP inhibition leads to sustained DNA damage that promotes the generation of tumoral neoantigens, programmed death-ligand 1 (PD-L1) expression on cancer cells, and immune cell infiltration, which might enhance responsiveness to ICIs. Preclinical data suggest that the combination ICIs with PARPi may result in a synergistic activity [113,114,115]. On this basis, the purpose of the ongoing Phase II PEMBROLA study (NCT05201612) is to evaluate the efficacy and safety of Pembrolizumab and Olaparib in patients with mCRC with HR deficiency [116]. Furthermore, in the neoadjuvant setting, a phase I/II trial (NCT04926324) is testing the combination of Dostarlimab (anti-PD-1) and Niraparib for up to 12 weeks after short-course radiotherapy in patients with locally advanced rectal cancer (LARC) [117].

## 5. Combinations of ICIs and New Immunotherapy Drugs

Combined blockade with immunotherapy strategies has been evaluated to overcome the inefficacy of the PD-1/PD-L1 axis alone in pMMR/MSS mCRC.

The CheckMate-142 study, a phase II trial of nivolumab with or without ipilimumab in MSI-H mCRC (NCT02060188), included 23 patients with refractory pMMR/MSS mCRC [118,119]. In this small cohort, only one PR was observed, with no signs of clinical activity in the remaining pMMR/MSS patients while median PFS was 1.4 months. These results showed how nivolumab/ipilimumab combination did not deserve further investigation in patients with pMMR/MSS mCRC due to limited clinical activity [2]. Similarly, the TAPUR phase II basket trial (NCT02693535) explored the anti-tumor activity of commercially available targeted agents in heavily pretreated patients with advanced cancers harboring genomic alterations. A small cohort of 10 pMMR/MS stable mCRC patients with high TMB, defined as ≥9 mutations/megabase (Muts/Mb), were treated with Nivolumab plus Ipilimumab induction followed by Nivolumab maintenance until disease progression was analyzed. The ORR was 10%, median PFS and OS 3.1 and 9.9 months, respectively. Authors concluded that Nivolumab in combination with Ipilimumab did not have sufficient clinical activity in patients with MSS and high-TMB mCRC for further evaluations [120].

Combination of durvalumab (anti-PD-L1) with tremelimumab (anti-CTLA-4) was assessed in randomized phase II trial CCTG CO.26 in pretreated patients.

A total of 180 patients were randomized in a 2:1 ratio to receive tremelimumab 75 mg every 4 weeks for first four cycles, and durvalumab 1500 mg every 4 weeks plus best supportive care (BSC), or best supportive care alone. Of 179 treated patients, 166 had MSS tumors. OS, the primary endpoint, was 6.6 months in the experimental group versus 4.1 months with BSC alone. No differences in PFS were shown. It is important to remark that correlative analysis revealed that patients with plasma TMB of 28 or more variants per megabase (21% of MSS tumors) had the greatest OS benefit from the combined Immune Checkpoint Inhibition (hazard ratio [HR]: 0.34, *p* = 0.004). Therefore, CCTG CO.26 is the first study to suggest that the combination of ICIs (anti-PD-L1 plus anti-CTLA-4) may prolong OS in patients with MSS-advanced CRC, especially in those with a high TMB [121].

Botensilimab is a novel innate/adaptive immune activator anti-CTLA-4 which promotes intratumoral regulatory T cell depletion via enhanced Fc-gamma receptor signaling and activation on natural killer cells and macrophages. It also favors optimized T cell priming, activation, and memory formation by enhancing antigen-presenting cell/T cell co-engagement. Further, Botensilimab has been developed to reduce complement fixation and complement-mediated toxicities. NCT03860272 is the first trial of botensilimab plus or minus the anti-PD-1 balstilimab in patients with immunotherapy-refractory metastatic solid tumors. In the expanded phase IA/B study a cohort of 41 evaluable, heavily pretreated MSS mCRC patients received botensilimab at 1 or 2 mg/kg every 6 weeks plus balstilimab 3 mg/kg every 2 weeks. The combination regimen demonstrated remarkable activity with an ORR of 24% (10/41) and a DCR of 73% (30/41). Interestingly, the observation that ORR was higher (42%) in patients without liver metastases (or with resected or ablated liver metastases), suggests that the sites of metastases may be predictive of response to immunotherapy [122].

Promising results came from new ICIs that block the lymphocyte activation gene-3 (LAG-3) in combination with PD-1 blockade. LAG-3 is a negative costimulatory surface molecule, expressed by immunity cells, which is involved in inhibiting effector T cell proliferation and activation, and in enhancing regulatory T cell suppressor activity [123]. LAG-3 acts synergistically with PD-1 to suppress antitumor immunity and is co-expressed with PD-1 on anergic T cells. Preclinical studies showed that the dual blockade of LAG-3 and PD-1 potentiates effector T cell activity and results in reversal of T-cell anergy [124,125,126,127]. In the phase III RELATIVITY-047 study of fixed-dose combination of the anti-LAG-3 antibody relatlimab and nivolumab versus nivolumab alone in patients with advanced and untreated melanoma, PFS was statistically higher in the experimental arm with the dual blockade (median 10.1 months versus 4.6 months; hazard ratio 0.75, 95% confidence interval (CI) 0.6–0.9; *p* = 0.0055) [128]. The combination of Nivolumab and Relatlimab in patients with MSS mCRC is being tested in an ongoing phase II trial (NCT03642067), whereas the association of Nivolumab-Relatlinib fixed dose versus Regorafenib or TAS-102 in chemorefractory mCRC is under investigation in the phase III randomized RELATIVITY-123 trial (NCT05328908) [129,130]. Results from the dose confirmation phase I first-in-human clinical trial testing the anti-LAG-3 antibody Favezelimab plus Pembrolizumab in previously treated patients with MSS mCRC showed that the dual blockade had a manageable safety profile and a promising antitumor activity, particularly in PD-L1 CPS ≥ 1 tumor. A total of 89 patients received the combined blockade. ORR was 6.3%, Median PFS was 2.1 months and median OS was 8.3 months. An exploratory analysis demonstrated that tumors with a PD-L1 combined positive score CPS ≥ 1 had a higher ORR (11.1%) and a longer OS (12.7 months) compared to the CPS < 1 group (ORR 2.9%, OS 6.7 months) [131]. 

Maraviroc is an antagonist of C–C motif chemokine receptor 5 (CCR5) that leads to a repolarization of macrophages toward an M1-like phenotype promoting a tumor-inhibiting immune milieu. In the PICASSO phase 1 trial, the association of Pembrolizumab and Maraviroc in 19 evaluable patients with pMMR mCRC showed an ORR of 5.3%, a median PFS of 2.1 months, and a median OS of 9.8 months [132]. Results from the phase Ib trial testing the activator of TLR9 Pixatimod plus nivoumab in heavily pretreated patients with MSS mCRC, metastatic pancreatic ductal adenocarcinoma and other solid tumors were recently reported (NCT05061017). Of 33 patients enrolled in the MSS mCRC cohort, 25 participants were evaluable. The maximum tolerated dose of pixatimod—25 mg—in combination with 240 mg nivolumab was well-tolerated and the DCR was 44%, representing an interesting signal of efficacy in a heavily pretreated refractory patient population [133]. The aim of the ongoing phase I/II STOPTRAFFIC-1 trial (NCT04599140) is to evaluate the combination of Nivolumab and SX-682, an oral small-molecule inhibitor of the CXCR1/2 chemokine receptors involved in MDSC-recruitment to tumors in RAS-Mutated and MSS mCRC [134]. Furthermore, the association of Pembrolizumab and the novel immunomedicine NC410, a monoclonal antibody which blocks the inhibitory receptor LAIR-1 (Leukocyte-associated immunoglobulin-like receptor 1) expressed on T cells, B cells, and NK cells, is under investigation in a recruiting phase Ib/2 trial (NCT05572684) [135]. 

## 6. Combinations of ICIs with Radiotherapy

Radiotherapy determines pro-immune effects by damaging DNA of cancer cells that results in an increased expression of tumor-associated antigen presentation, T cells recruitment and activation, and up-regulation of inflammatory cytokines. In addition to the local therapeutic effect, radiotherapy may act distally through the so-called “abscopal effect” which is promoted by the activation of the immune system against cancer cells and consists in a shrinkage of metastatic sites distant from the field of irradiation [136,137,138]. The combination of radiotherapy and ICIs has been shown to be effective and safe in the treatment of non-small-cell lung cancer and melanoma, due to their own immunogenic biology [139,140]. 

However, despite the strong rationale, the role of radiotherapy in combination with ICIs in MSS mCRC remains controversial, due to limited evidence of efficacy. A phase 2 single-arm study (NCT03122509) of durvalumab and tremelimumab with concurrent radiotherapy in pMMR mCRC patients who had received at least two prior lines of chemotherapy showed modest outcomes with an ORR of 8.3% (2/24 patients), median PFS 1.8 months and median OS 11.4 months [141]. Similarly, NCT02888743 was a randomized phase II trial designed to investigate the safety and activity, in terms of ORR, of durvalumab and tremelimumab with or without high-dose or low-dose radiation therapy in patients with mCRC or non-small-cell lung cancer. A preliminary analysis of the colorectal cohort demonstrated no significant radiotherapy-related toxicities. However, the best response of stable disease in one patient did not support the use and further investigation of this regimen, due to insufficient clinical activity [142]. The results of another phase II trial (NSABP FC-9) with a combination of durvalumab plus tremelimumab following palliative hypofractionated radiation in patients with MSS mCRC progressing on chemotherapy (NCT03007407) are still awaited [143]. 

A phase 2 trial (NCT03104439) investigated the combination of hypofractionated radiotherapy (8 Gy in three fractions to a single metastatic lesion) with ipilimumab and nivolumab in MSS and MSI-H mCRC and pancreatic cancer. A total of 40 patients with mCRC were enrolled and treated. In the colorectal cohort, the disease control rate was 25% with ORR of 10%. Median PFS was 2.4 months and median OS 7.1 months. DCR and ORR were higher among the 27 patients who received radiation therapy (37% and 15%, respectively) [144]. Similarly, the association of ipilimumab and nivolumab with radiotherapy in MSS mCRC is under investigation in a recruiting phase 2 trial (NCT04575922) [145]. Instead, the aim of the NCT02437071 ongoing phase 2 trial is to assess efficacy and safety of Pembrolizumab single-agent in association with radiotherapy or ablation in mCRC patients regardless of microsatellite status [146]. Moreover, another ongoing phase 2 trial, NCT05160727, is investigating the radiosensitization effects of Tislelizumab and irinotecan in mCRC patients [147].

The association of neoadjuvant chemoradiation and ICIs in locally advanced rectal cancer (LARC) has already been tested in several trials such as the Lebanese AVERECTAL trial, the Italian AVANA trial, and the Japanese VOLTAGE trial, with encouraging pCR rates of 38%, 23%, and 33%, respectively [148,149,150]. Moreover, lots of trials on the combination of ICIs and radiotherapy plus or minus chemotherapy are ongoing in the neoadjuvant setting for treating LARC [151]. 

## 7. Immunohistochemical and Circulating Biomarkers of Response to ICIs in MSS/pMMR mCRC 

MSS mCRC sensitivity to immunotherapy must actually be at the center of the discussion, in order to get the best benefits from these therapies. Immunohistochemical biomarkers beyond the MMR status might be essential to differentiate the MSS populations and discover new and promising pathways and strategies to get the proper immunotherapies for specific subgroups of patients. 

PD-L1 is an immunohistochemical biomarker frequently used in clinical practice for many cancer histotypes. PD-1 is expressed on the surface of T cells, B cells, and natural killer cells and the ligand, PD-L1, has been found to be expressed on the surface of tumor cells to a higher or lower degree. Upon ligating its receptor, PD-L1 has been reported to decrease T cell receptor-mediated proliferation and cytokine production. Thus, PD-L1 might play an important role in tumor immune evasion [152]. Combined Positive Score (CPS) related with PD-L1 expression is widely used by pathologists and oncologists to select patients for immunotherapy in various types of cancers. MSI-H mCRC have increased PD-L1 expression (56% vs. 21% in MSS tumors; *p* = 0.007) [153]. Thus far, in mCRC, regardless of MMR status, PD-L1 expression has not proved to be predictive of response to immunotherapy. For example, Overman et al., in the subgroup analysis of a multicenter, open-label, phase 2 trial, in which patients with MSI-H/dMMR refractory mCRC were treated with single-agent nivolumab, observed no significant differences between PD-L1 < 1% and >1% tumors [118]. Llosa et al., observed a scarce PD-L1 expression in CRC cell lines, unlike the high predominance seen in the surrounding myeloid cells; this aspect is totally different from other more immunogenic tumors, such as renal or lung cancer. This is probably the real reason why PD-L1 is not a good ICH biomarker for CRC [154].

Tumor mutational burden (TMB) is another noteworthy biomarker that may predict the response to ICIs. TMB represents the number of mutations per megabase (Mut/Mb) of DNA that were sequenced in a specific cancer. It is supposed that a high TMB may be correlated with the cancer neoantigen burden that induces immunogenicity. With a higher number of mutations detected, and consequentially an increase in the number of neo-epitopes, it is more likely that one or more of those neoantigens could be immunogenic and trigger a T cell response. Initially, TMB was identified as a biomarker for ICIs in melanoma, and subsequent studies suggested a possible clinical role for TMB in non-small-cell lung cancer. We actually lack analytics and the most effective technique.. The next-generation sequencing (NGS) techniques, the whole genome sequencing (WGS), or whole exome sequencing (WES) include the entire coding regions and has been considered the reference standard [155]. Currently, there is no TMB cutoff [156]. KEYNOTE-158 demonstrated activity of pembrolizumab in patients with TMB-high tumors, showing an ORR = 28.3% (ORR = 24.8% in TMB-H/non-MSI-H tumors). On the basis of this result, the US FDA approved the use of pembrolizumab for TMB-H solid tumors on 16 June 2020 [157]. In particular, Pembrolizumab can be used in pretreated mCRC with TMB ≥ 10 mut/Mb, a subgroup with an ORR of 29.4% compared to 6.3% in TMB < 10 mut/Mb. Foundation One and MSK-IMPACT panels have been approved by the FDA for the assessment of TMB. Nevertheless, only a small percentage of MSS tumors are estimated to be TMB-H (about 3%) [155,156,158,159]. However, the clinical implications of this universal cutoff of TMB ≥ 10 for patients with MSS mCRC remain debatable [160]. Goodman et al., with an analysis of 21 types of cancer, including gastrointestinal tumors (*n* = 151), showed that TMB was significantly associated with the ORR and survival prognosis of PD-1/PD-L1 inhibitors. Patients with high TMB (≥20 mutation/Mb) had significantly better responses (ORR: 58% vs. 20%, *p* = 0.0001), longer median PFS (12.8 months vs. 3.3 months, *p* ≤ 0.0001), and improved median OS (not reached vs. 16.3 months, *p* = 0.0036) than patients with moderate or low TMB (<19 mutation/Mb) [161]. 

MyPathway was a phase IIa basket study, which evaluated atezolizumab in advanced solid tumors with TMB ≥ 10 mut/Mb. The study included 19 tumor types, including mCRC, and showed a promising clinical activity of immunotherapy patients with TMB ≥ 16 mut/Mb. In particular, ORR was 38.1% and DCR 61.9% in this setting vs. 2.1% and 22.9% in TMB < 16 mut/Mb. In the cohort of patients with mCRC (*n* = 21), including both MSI and MSS patients, an ORR of 70% was observed in 10 patients with TMB ≥ 16 mut/Mb. Interestingly, an objective response was achieved in three out of five patients with MSS tumors and TMB ≥ 16 mut/Mb, suggesting a role for TMB as a predictor of benefit from ICIs in MSS mCRC, but also in MSI mCRC patients who may not benefit from ICIs [162]. More studies are still necessary to determine and define the optimal cut-off for TMB in this setting. 

About 2% of all CRC tumors have somatic or germline mutations in the POLE and POLD1 genes. POLE and POLD1 have a key role in DNA proofreading and replication and mutations in their genes, in particular those involving the POLE exonuclease domain, lead to a hypermutated phenotype without dMMR expression [163,164]. A high TMB, with an average of 158 Mut/Mb, and an increase of TILS is observed in these tumors. A study by Keshinro A et al. on 499 CRC cases, including 11 POLE/POLD1-mutated tumors, showed higher rates of TILs in POLE- and POLD-mutated tumors (82%) in comparison with MSI-H (68%) and non-MSI-H CRC (4.5%) [165]. In a large retrospective analysis, Garmezy et al. confirmed the role of POLE and POLD1 pathogenic mutations as a predictive factor of response to ICIs irrespective of MMR status [166]. A phase II study is currently evaluating ICIs therapy with Durvalumab in MSI or POLE-mutated mCRC with ORR as primary endpoint [167], while another trial is currently testing nivolumab vs. nivolumab plus ipilimumab in the same setting [168].

Increasing evidence suggests a close correlation between the tumor growth and the microenvironment in which it develops, and particularly the lymphocyte populations. In this context, Immunoscore has been developed as a tumor-agnostic method to define both the prognosis and the immunogenicity of tumors, and might represent a predictive marker of response to immunotherapy. The Immunoscore is based on the quantification of lymphocyte populations, in particular CD3 and CD8-positive T cells, both at the tumor center and at the invasive margin. CD3- and CD8-immunostained formalin-fixed, paraffin-embedded slides are scanned and the two corresponding digital images validated by the operator. Image analysis is performed via a dedicated software (Immunoscore Analyzer, HalioDx) with an automatic detection. The classification goes from low densities, Immunoscore 0, to high densities classed as Immunoscore 4 (I4). Galon et al. observed a correlation between increasing score and longer patient survival [169]. Such prognostic superiority was shown to be statistically significant for stage I, II, and III CRC, with tumor progression and invasion depending on the immune parameters [170]. These immune infiltrates were also associated with better outcomes and a decreased probability of developing metastases [171]. Recently, Immunoscore was also proposed as a predictive biomarker for patients treated with immunotherapy. In particular, a subgroup analysis of Atezotribe evaluated the immunoscore (DetermaIO) as a predictive biomarker of the efficacy of Atezolizumab added to standard chemotherapy. DetermaIO was successfully determined in 122 of 218 enrolled patients, and 23 (27%) tumors were positive. Positive tumors achieved higher PFS benefit from an atezolizumab arm than negative ones (HR: 0.39 vs. 0.83), and a similar trend was observed in pMMR tumors (HR: 0.47 vs. 0.93). These preliminary results suggest a role for DetermaIO in the prediction of the benefit of adding atezolizumab to first-line FOLFOXIRI plus bevacizumab in mCRC [172]. Clearly, a validation of the test in larger series is necessary to justify its use in clinical practice. 

Inflammatory cytokines such as tumor necrosis factor-α (TNF-α), Interferon-γ (IFN-γ), Interleukins such as IL-6 or IL-8, and transforming growth factor-β (TGF-β), have an important role during CRC development and progression. In particular, their activity in the regulation of tumor microenvironment (TME) and immune cells differentiation suggest a possible role as biomarkers for prediction of ICIs efficacy. IL-6 is a key regulator of inflammation via the JAK/STAT signaling pathway and IL-6 levels are associated with tumor stage, metastasis, and survival in CRC [173]. Interestingly, in a study of 209 resected CRC samples, high IL-6 expression in tumor-infiltrating immune cells was associated with accumulation of immunosuppressive cells in the TME, such as MDSCs and Treg [174]. Several studies have evaluated the association between IL-6 levels and ICIs response in particular in non-small-cell lung cancer and melanoma, while data in pMMR CRC are lacking. Overall, these studies showed that higher baseline IL-6 levels were associated with shorter survival while its increase during ICI treatment was associated with tumor progression, suggesting a role of IL-6 as a negative biomarker for immunotherapy [175]. TNF-α is another key regulator of inflammatory response, and preclinical studies suggest that it may act as a negative biomarker for ICI treatment through upregulation of PD-L1 on tumor cells and T cells [176] as confirmed by a small study on melanoma patients treated with nivolumab [177]. IL-8, also known as CXCL8, has immune-modulatory effects that have been investigated in different malignancies, including CRC. In particular, a preclinical study showed a critical role for CXCL8 in promoting M2 macrophage polarization and inhibition of CD8+ T cell infiltration, thus contributing to the emergence of an immunosuppressive microenvironment [178]. The predictive role of inflammatory cytokines in mCRC patients treated with ICIs is worthy of further clinical evaluation in prospective trials. 

## 8. Discussion

Despite many advances in understanding the biology of mCRC, several obstacles are still standing against successful treatment of this tumor with ICIs. 

The first point is that, speaking about immunotherapy in mCRC, we have to consider two completely different subgroups of patients. In fact, dMMR/MSI-H CRC are genetically unstable, and gather a high mutational burden (TMB > 12 mutations per megabase) and highly immunogenic neoantigens, and this explains the high efficacy of ICIs in these patients. Moreover, a shorter PFS with first-line chemotherapy and a trend to better OS with anti-VEGF vs. anti-EGFR agents has been observed [179]. On the contrary, MSS/pMMR CRC are characterized by chromosomal instability, aneuploidy, a much lower TMB (<8.24 mutations per megabase), and resistance to ICIs [180,181]. For these reasons, in all patients diagnosed with mCRC, the screening of MSI status is recommended as an essential step in order to select those who would benefit from immunotherapy. ICIs have shown high efficacy in MSI-H mCRC, with a current indication of pembrolizumab as the standard of care in first and later lines, and nivolumab plus ipilimumab in later lines. However, the use of ICIs as monotherapy has shown no benefit in MSS mCRC, which account for about 95% of patients. 

Undoubtedly, the immunosuppressive function of the TME is one of the principal causes of resistance to immunotherapy in these tumors. In particular, the absence of an effective immune infiltrate in MSS tumors, along with a considerably lower mutation load, are the basis of the low immunogenicity of these tumors. Due to these reasons, combination strategies and novel agents are needed in order to increase the mutational load and modulate the immunosuppressant TME. Different strategies are currently under investigation to overcome resistance so as to expand the application scope of immunotherapy: the combination with chemotherapy, in particular oxaliplatin-based regimens, as well as bevacizumab and anti-EGFR inhibitors, have produced promising results, to be validated in randomized trials. Also, Temozolomide has been shown to induce a transient MSI status in silenced MGMT MSS mCRC, which may lead to sensitization to ICIs. The combination of target therapies, such as KRAS/BRAF, MEK, PIK3CA/AKT/mTOR inhibitors or Multitarget TKIs, constitutes another interesting line of research. However, data from phase 1 or 2 studies require validation in larger case series to transform the strong biological rationale into evidence capable of modifying clinical practice. Radiotherapy, through both its pro-immune and ‘abscopal’ effects, is another potential enhancer of immunosensitivity, even if results in mCRC are lacking, and the application to LARC seems more attractive with respect to metastatic disease. Also, the combination with other ICIs is promising, although it has not yet demonstrated a significant improvement to patients’ outcomes compared to monotherapy.

As previously said, there is a need of reliable biomarkers of efficacy for immunotherapy beyond the MMR status. Differently from other malignancies, PD-L1 expression is not correlated with ICI efficacy in CRC. On the other hand, TMB might have a relevant role in the selection of MSS patients who could benefit from ICIs, even if its role in CRC is not yet clear. Currently, Immunoscore seems the most promising biomarker, even if its role has been demonstrated only in early-stage disease and in Atezotribe trials. 

## 9. Conclusions

The majority (about 95%) of mCRC cases are pMMR or MSS, and unfortunately, this condition is associated with an inherent resistance to ICIs. However, several trials are investigating the right way to overcome resistance and/or to define subgroups of patients responsive to ICIs. A new era seems to be emerging. In particular, the association of polychemotherapy and biological agents, the sequential treatment with temozolamide, and the selection of patients with immunohistochemical markers, appear to be of possible short-term application. Moreover, a better understanding of the TME and the tumor immune escape mechanisms, as well as larger prospective randomized clinical trials using different combination modalities, will help obtain further improvement.

## Figures and Tables

**Figure 1 cancers-15-05189-f001:**
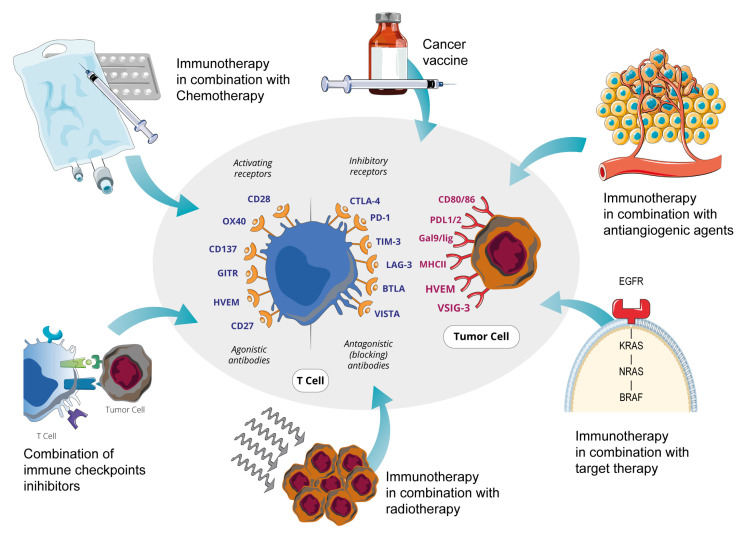
T-cell and tumor cell interaction mediated by activating or inhibitory immunological checkpoints and potential combination therapies to enhance efficacy. Agonistic or antagonist antibodies, by activating or inhibiting immunological checkpoints, lead to T-cell activation and expansion, and promote antitumor immune response. Combination strategies may enhance immunotherapy efficacy. OX40 or TNFRSF4: tumor necrosis factor (ligand) superfamily member 4; GITR: Glucocorticoid-Induced TNFR-Related; HVEM: Herpes Virus Entry Mediator; CTLA4: Cytotoxic T Lymphocyte Antigen-4; PD-1: programmed cell death-1; PD-L1: programmed cell death-1 ligand 1; TIM3: T cell immunoglobulin and mucin domain-containing protein 3; LAG-3: Lymphocyte Activation Gene-3 Protein; BTLA: B and T lymphocyte attenuator; VISTA: V-domain immunoglobulin suppressor of T cell activation; Gal9: Galectin-9; MHC: Major Histocompatibility Complex; VSIG-3: V-set and Ig domain-containing 3; Car-T: Chimeric Antigens Receptor T Cells; TIL: tumor infiltrating lymphocytes; Car-NK: Chimeric Antigens Receptor-Natural Killer Cells.

**Table 1 cancers-15-05189-t001:** Clinical trials with ICIs in dMMR/MSI-H mCRC patients.

Study	Treatment	Phase	Setting & Study Population	Sample Size (*n* pts)	End-Points	Results
**ICIs in dMMR/MSI mCRC**						
Keynote 016 NCT01876511 (2015)	Pembrolizumab	II	≥Second line MSI-H/dMMR	11	1: ORR, 20 week PFS 2: mPFS and mOS	ORR 40% 20 week PFS 78% mPFS and mOS not reached
Keynote 177 NCT02563002 (2020)	Pembrolizumab vs chemotherapy	III	First line MSI-H/dMMR	307	1: PFS, OS 2: ORR, safety	mPFS 16.5 vs. 8.2 mo; HR, 0.60; *p* = 0.0002. OS 13.7 vs. 10.8 mo. ORR: 43.8 vs. 33.1%,
CheckMate 142 NCT02060188 (2022)	Nivolumab plus Ipilimumab	II	First line MSI-H/dMMR	45	1: ORR 2: DCR, PFS, OS, safety	ORR 69% DCR 84% mPFS and mOS not reached
CheckMate 142 NCT02060188 (2022)	Nivolumab plus Ipilimumab	II	≥Second line MSI-H/dMMR	119	1: ORR 2: DCR, PFS, OS, safety	ORR 65% DCR 81% mPFS and mOS not reached
CheckMate 142 NCT02060188 (2022)	Nivolumab	II	≥Second line MSI-H/dMMR	74	1: ORR 2: DCR, PFS, OS, safety	ORR 39% DCR 69% mPFS and mOS not reached
**ICIs in pMMR/MSS mCRC**						
Keynote 016 NCT01876511 (2015)	Pembrolizumab	II	≥Second line MSS/pMMR	21	1: ORR, 20 week PFS 2: mPFS and mOS	ORR 0% 20 week PFS 11% mPFS 2.2 mo mOS 5.0 mo
IMblaze 370 NCT02788279 (2019)	Atezolizumab	III	≥Third line MSS/pMMR (cohort B)	90	1: OS 2: PFS; ORR	mOS 7.1 mo mPFS 1.94 mo ORR 2%

**Table 2 cancers-15-05189-t002:** Clinical trials investigating ICIs in combination with chemotherapy in MSS/pMMR mCRC.

Study	Treatment	Year	Phase	Setting & Study Population	Sample Size (*n* pts)	End-Points	Results
	**ICI in combination with chemotherapy**						
NCT02375672	FOLFOX + Pembrolizumab	2017	II	First line MS NS	30	1: PFS 2: ORR, DCR	mPFS not reached ORR: 53%, DCR: 100%
MEDETREME NCT03202758	FOLFOX + Durvalumab + Tremelimumab followed by Durvalumab	2021	Ib/II	First line MS NS	57	1: 6 mo PFS rate 2: ORR, DCR, OS, safety	6 mo PFS 63.2% 12 mo PFS 39% ORR 61%, DCR 89%
NCT02860546	FTD/TPI + Nivolumab	2021	II	Refractory MSS	18	1: irORR 2: PFS, DCR, ORR, OS, safety	irORR not reached mPFS: 2.8 mos
	**ICI in combination with chemotherapy and anti-VEGF agents**						
MODUL cohort 2 NCT02291289	FOLFOX + BEV followed by FP + BEV vs. FP + BEV + Atezolizumab	2018	II	First line MS NS	445 (297 vs. 148)	1: PFS 2: OS	PFS: 7.4 mos vs. 7.2 mos, HR 0.96, *p* = 0.72 OS: 51%, HR = 0.86, *p* = 0.28
ATEZOTRIBE NCT03721653	FOLFOXIRI + BEV + Atezolizumab followed by FP + BEV + Atezolizumab vs. FOLFOXIRI + BEV followed by FP + BEV	2022	II	First line MS NS	218 (145 vs. 73)	1: PFS 2: ORR, irORR, R0 resection rate	mPFS: 13.1 mos vs. 11.5 mos HR 0.69, *p* = 0.012
NIVACOR NCT04072198	FOLFOXIRI + BEV + Nivolumab followed by BEV + Nivolumab vs. FOLFOXIRI + BEV followed by BEV	NP	II	First line MS NS	Recruiting	1: ORR 2: OS, TTP, DoR, safety	NA
CA2099X8 NCT03414983	FOLFOX + BEV + Nivolumab vs. FOLFOX + BEV	2022	II	First line MS NS	195 (127 vs. 28)	1: PFS 2: ORR, DCR, TTR, DoR, OS, safety	mPFS: 11.9mos vs. 11.9 mos HR 0.81, *p* = 0.30 18 mo PFS: 28% vs. 9% ORR 76% vs. 31%
BACCI NCT02873195	Capecitabine + BEV + Atezolizumab vs. Capecitabine + BEV	2019	II	Refractory MS NS	128 (82 vs. 46)	1: PFS 2: 12 mo OS rate	mPFS: 4.4 mos vs. 3.6 mos. HR 0.75, *p* = 0.07 12 mo OS: 44.5% vs. 42%
COLUMBIA-1 NCT04068610	FOLFOX + BEV + Durvalumab + Oleclumab vs. FOLFOX + BEV	NP	Ib/II	First line MSS	Recruiting	1: safety, ORR 2: DoR, DCR, PFS, OS, safety	NA
POCHI NCT04262687	CAPOX + BEV + Pembrolizumab	NP	II	First line MSS	Recruiting	1: 10 mos PFS 2: OS	NA
3B-FOLFOX NCT05627635	FOLFOX + BEV + Botensilimab + Balstilimab	NP	I/II	from 1st to 3rd line MSS	Active from 25 November 2022, not yet recruiting	1: safety (phase 1), ORR (phase 2) 2: ORR, PFS e OS (phase 1), PFS, OS, DoR (phase2)	NA
NCT05314101	FTD/TPI + Bevacizumab + Tislelizumab	NP	II	Refractory MSS	Recruiting	1: PFS 2: ORR, OS,	NA
	**ICI in combination with chemotherapy and anti-EGFR agents**						
AVETUX NCT03174405	FOLFOX + Cetuximab + Avelumab	2020	II	First line RAS/BRAF WT MS NS	43	1: 12 mos PFS 2: ORR, OS, safaty	12 mos PFS 40% ORR 79.5%, DCR 92.3%, mPFS 11.1 mos
FIRE-6	FOLFIRI + Cetuximab + Avelumab	NP	II	First line1 RAS WT MS NS	Recruiting	1: PFS 2: ORR, OS, safety	NA
AVETUXIRI NCT03608046	IRINOTECAN + Cetuximab + Avelumab	2021	II	Refractory BRAF WT MSS	23 (10 cohort A RASwt, 13 cohort B RASmut) at interim analysis, Recruiting	1: irORR, safety 2: DCR, PFS, OS	Interim analysis irORR 30% (RASwt), NR in cohort B (RAS mut) 6 mos PFS 40% (RASwt) vs. 38.5% (RAS mut) 12 mos OS 50% (RASwt) vs. 46.2% (RASmut) mPFS 4.2 vs. 3.8 mos mOS 12.7 vs. 14 mos
CAVE NCT04561336	Cetuximab + Avelumab	2022	II	Refractory RAS WT MS NS	77	1: OS 2: PFS, ORR, safety	mOS 11.6 mos mPFS 3.6 mos
CAVE2 NCT05291156	Catuximab + Avelumab vs. Cetuximab	NP	II	Refractory RAS/BRAF WT MS NS	Recruiting	1: OS 2: PFS, ORR, DCR, safety	NA
AVETRIC NCT04513951	FOLFOXIRI + Cetuximab + Avelumab followed by 5FU + Cetuximab + Avelumab	NP	II	First line RAS/BRAF WT MS NS	Recruiting	1: PFS 2: ORR, OS, irORR, safety, R0 resection rate	NA
	**ICI in combination with Temozolomide**						
MAYA	Temozolomide followed by ipilimumab and Nivolumab	2022	II	Refractory MGMT silenced mCRC MSS	33	1: PFS 2: OS, ORR, DoR, safety	8-months PFS 36% mPFS 7.0 mos mOS 18.4 mos
ARETHUSA	Temozolomide followed by pembrolizumab	2022	II	Refractory MGMT silenced mCRC MSS	21	1: ORR 2: PFS, OS, safety	Interim analysis DCR 67%

Abbreviations: MSS, microsatellite stable; NR, not reached; NA, not available; NS, not specified; NP, not published; ORR, overall response rate; OS, overall survival; PFS, progression-free survival; BEV, Bevacizumab; DCR, Disease Control Rate; EGFR, Epidermal Growth Factor Receptor; FOLFOX, 5-FU + Folinic Acid + Oxaliplatin; FP, Fluoropyrimidine; HR, Hazard Ratio; irORR, immune-related Overall Response Rate; mCRC, Metastatic Colorectal Cancer; mos, months; VEGF, Vascular Endothelial Growth Factor. MS NS, Microsatellite Status non specified; WT, wild type.

**Table 3 cancers-15-05189-t003:** Clinical trials investigating ICIs in combination with target therapy.

Study (Name and/or NCT)	Treatment	Phase	Setting & Study Population	Sample Size (*n* pts)	End-Points	Results
**ICI in combinations with KRAS G12C inhibitors**						
Codebreak101	Sotorasib + AMG404	Ib	Refractory KRAS G12C MT MSS	Recruiting	1: safety 2: DoR, DCR, PFS	NA
**ICI in combination with BRAF/MEK inhibitors**						
NCT04017650	Encorafenib + Cetuximab + Nivolumab	I/II	Refractory BRAF V600E MT MSS	26	1: ORR, safety	ORR:50% DCR: 96% mPFS: 7.4 mos mOS: 15.1 mos
NCT01988896	Cobimetinib + Atezolizumab	Ib	Refractory	23	1: DLT 2: ORR, PFS, OS	ORR: 17%
NCT03668431	Dabrafenib + Trametinib + Spartalizumab	II	Refractory BRAF V600E MT MS NS	37	1: ORR 2: PFS, OS, DoR, scRNAseq	ORR: 24.3% mPFS: 4.3 mos mOS: 13.6 mos
**ICI in combination with PI3K/AKT/mTOR**						
NCT03711058	Copanlisib + Nivolumab	I/II	Refractory mCRC MSS	Recruting	1: DLT, ORR 2: DCR, DOR, PFS; OS	NA
**ICI in combination with multitarget TKIs**						
IMblaze370	Atezolizumab +/− cobimetinib vs. regorafenib	III	Refractory mCRC MS NS	363 (183 vs. 90 vs. 90)	1: OS 2: ORR, DoR, PFS	mOS: 8.9 mos vs. 7.1 mos vs. 8.5 mos, HR 1, *p* = 0.99
REGONIVO	Regorafenib + Nivolumab	Ib	Refractory mCRC MS NS	25 (CRC cohort)	1: DLT 2: AEs, ORR, DCR, PFS, OS	ORR: 36%, PFS: 7.9 mos mOS: NR
NCT04126733	Regorafenib + Nivolumab	II	Refractory mCRC MSS	94	1: ORR 2: AEs, DoR, PFS, OS	ORR: 7% mPFS: 1.8 mos mOS: 11.9 mos
REGOMUNE	Regorafenib + Avelumab	II	Refractory mCRC MSS	48	1:ORR 2: PFS, OS, safety	ORR: 0% mPFS = 3.6 mos mOS = 10.8 mos
CAMILLA	Cabozantinib + Durvalumab	I/II	Refractory mCRC MSS	20 (CRC cohort)	1: DLT 2: ORR, DCR, PFS, OS	ORR: 23.5% DCR: 88.2% mPFS: 4.6mos mOS: 9.6 mos

Abbreviations: ICIs: Immune checkpoint inhibitors; MSS: microsatellite stable; NR: not reached; NA: not available or not announced; NS: not specified; ORR: overall response rate; OS: overall survival; PFS: progression-free survival; DCR: Disease Control Rate; HR: Hazard Ration; mCRC: Metastatic Colorectal Cancer; mos: months; MS NS: Microsatellite Status non-specified; MT: mutant.

## Data Availability

The datasets generated and/or analyzed during the current study are available from the corresponding author on reasonable request.

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
