# Peer review of "Immunocheckpoint Inhibitors in Microsatellite-Stable or Proficient Mismatch Repair Metastatic Colorectal Cancer: Are We Entering a New Era?"

_cancers, 2023, doi:10.3390/cancers15215189_

Round 1

Reviewer 1 Report

Comments and Suggestions for Authors

This review article was written for immune checkpoint inhibitors in MSS or pMMR metastatic colorectal cancer. Although a same review was published in the same journal 2 years ago, the volume and references of this article was increased compare to previous article. Hence, this article might be published in this Journal. Unfortunately, some concerns should be resolved.

Major comments

1. The title and aim of this article is to explain clearly the perspective of immunotherapy for MSS colorectal cancer. Unfortunately, there is no clear description on the “new era” or present status on immunotherapy for MSS CRC. The authors should summarize and list up the novel advances and perspective, point to point, for the future expected for MSS CRC immunotherapy. This article contains just the list of contents of literatures. Please describe clearly the novel advances and perspective, in Abstract or Simple summary or Conclusion. Now, there are same description repeated in there three components. The authors should change description.

2. The contents of Figure 1 is poor for the review. The high-resolution image should be made for Figure 1. And also, Figure 1 should graphically-demonstrate the concept of new era of immunotherapy for MSS CRC, not only characters but also graphics. 

Author Response

Reviewer 1

This review article was written for immune checkpoint inhibitors in MSS or pMMR metastatic colorectal cancer. Although a same review was published in the same journal 2 years ago, the volume and references of this article was increased compare to previous article. Hence, this article might be published in this Journal. Unfortunately, some concerns should be resolved.

 Major comments

  1. The title and aim of this article is to explain clearly the perspective of immunotherapy for MSS colorectal cancer. Unfortunately, there is no clear description on the “new era” or present status on immunotherapy for MSS CRC. The authors should summarize and list up the novel advances and perspective, point to point, for the future expected for MSS CRC immunotherapy. This article contains just the list of contents of literatures. Please describe clearly the novel advances and perspective, in Abstract or Simple summary or Conclusion. Now, there are same description repeated in there three components. The authors should change description.

Reply: Thank you for the comment. We have changed both the simple summary and the conclusion, specifying what’s most promising in the near future for MSS CRC immunotherapy. The abstract was not changed.

  1. The contents of Figure 1 is poor for the review. The high-resolution image should be made for Figure 1. And also, Figure 1 should graphically-demonstrate the concept of new era of immunotherapy for MSS CRC, not only characters but also graphics.

Reply: Thank you very much for this suggestion. We have perfected the image quality and also added graphics about future combination strategies. We believe the message to readers is now much clearer and more direct.

Reviewer 2 Report

Comments and Suggestions for Authors

Good day! I have no additional questions for the authors

Author Response

Reviewer 2

Good day! I have no additional questions for the authors

Reply: Thank you for taking the time to review our manuscript.

Reviewer 3 Report

Comments and Suggestions for Authors

Immunocheckpoint Inhibitors in Microsatellite Stable or proficient Mismatch Repair metastatic colorectal cancer: are we entering a new era? Laura Matteucci1 , Alessandro Bittoni1,* , Graziana Gallo , Laura Ridolfi and Alessandro Passardi

Colorectal cancer (CRC) is the second cause of cancer-related deaths in Europe.  Approximately 25% of CRC patients are diagnosed at stage IV. Most patients with of mCRC cases are mismatch-repair-  proficient (pMMR) and microsatellite-stable (MSS).

The Authors reviewed  the strategies implemented to overcome ICI resistance and/or define subgroups of patients with MSS or pMMR mCRC who may benefit from  immunotherapy.

POSITIVE ASPETCS

The high prevalence of patients with MSS and pMMR give to the review a significant clinical importance.

The Authors underline that a combination of MSS and MMR implies resistance to ICI.

SUGGESTIONS

I would recommend if I am allowed to so the following suggestions.

1-Include two tables showing the results of large randomized prospective studies which demonstrate

a)      improved response and observed survival and disease free survival of ICI for patients with MMI and dMMR.

b)      No positive response to ICI for patients with MMS and pMMR.

2) In Table 1 list the year of publication of the papers to give a better idea of the temporal evolution.

Explain for clarification the meaning of different phases of clinical experimentation (Phase II etc etc).

3) Avoid the term Hot tumor and cold tumor, using other more scientifically sound definitions.

4)I suggest to add a subchapter about inflammation and efficiency of ICI in patients with CRC MMS and pMMR. Conceptually, the level of inflammation may contribute to the end results of ICI. The Authors analyze d lymphocyte count. I would suggest to add considerations about the prognostic value of inflammatory parameters including high serum and tumoral levels of inflammatory cytokines (IL1;IL6;TNFalfa etc etc).

Comments on the Quality of English Language

Need for minor improvement 

Author Response

Reviewer 3
Colorectal cancer (CRC) is the second cause of cancer-related deaths in Europe.  Approximately 25% of CRC patients are diagnosed at stage IV. Most patients with of mCRC cases are mismatch-repair-  proficient (pMMR) and microsatellite-stable (MSS). The Authors reviewed  the strategies implemented to overcome ICI resistance and/or define subgroups of patients with MSS or pMMR mCRC who may benefit from  immunotherapy.

POSITIVE ASPETCS

The high prevalence of patients with MSS and pMMR give to the review a significant clinical importance. The Authors underline that a combination of MSS and MMR implies resistance to ICI.

Reply: Thank you for the positive comments, we agree that this topic is highly significant.

SUGGESTIONS

I would recommend if I am allowed to so the following suggestions.

1-Include two tables showing the results of large randomized prospective studies which demonstrate

  1. a)      improved response and observed survival and disease free survival of ICI for patients with MSI and dMMR.
  2. b)      No positive response to ICI for patients with MSS and pMMR.

Reply: Thank you for the comment. We added 1 Table summarizing the trials that analyzed ICI’s monotherapy efficacy in both MSS/pMMR and MSI/dMMR tumors.

2) In Table 1 list the year of publication of the papers to give a better idea of the temporal evolution. Explain for clarification the meaning of different phases of clinical experimentation (Phase II etc etc).

Reply: Thank you for the suggestion. We added dates and the requested clarification in Table 1.

3) Avoid the term Hot tumor and cold tumor, using other more scientifically sound definitions.

Reply: as suggested we removed the terms ‘hot’ and ‘cold’ throughout the text.

4) I suggest to add a subchapter about inflammation and efficiency of ICI in patients with CRC MMS and pMMR. Conceptually, the level of inflammation may contribute to the end results of ICI. The Authors analyze d lymphocyte count. I would suggest to add considerations about the prognostic value of inflammatory parameters including high serum and tumoral levels of inflammatory cytokines (IL1;IL6;TNFalfa etc etc).

Reply: We considered this comment very interesting and we added a sub-chapter about prognostic role of inflammatory citokines (lines 722-745)

Round 2

Reviewer 1 Report

Comments and Suggestions for Authors

The manuscript has been revised accurately according to the reviewers’ comments.